# Accuracy Assessment of the 2D-FFT Method Based on Peak Detection of the Spectrum Magnitude at the Particular Frequencies Using the Lamb Wave Signals

**DOI:** 10.3390/s22186750

**Published:** 2022-09-07

**Authors:** Lina Draudvilienė, Asta Meškuotienė, Renaldas Raišutis, Olgirdas Tumšys, Lina Surgautė

**Affiliations:** 1Ultrasound Research Institute, Kaunas University of Technology, K. Baršausko St. 59, LT-51423 Kaunas, Lithuania; 2Metrology Institute, Kaunas University of Technology, LT-44249 Kaunas, Lithuania; 3Department of Electrical Power Systems, Faculty of Electrical and Electronics Engineering, Kaunas University of Technology, Studentų St. 48, LT-51367 Kaunas, Lithuania

**Keywords:** signal processing, 2D-FFT method, Lamb waves, phase velocity, dispersion curve, frequency, reliability, uncertainties, systematic error

## Abstract

The 2D-FFT is described as a traditional method for signal processing and analysis. Due to the possibility to determine the time and frequency (*t*,*f*) domains, such a method has a wide application in various industrial fields. Using that method, the obtained results are presented in images only; thus, for the extraction of quantitative values of phase velocities, additional algorithms should be used. In this work, the 2D-FFT method is presented, which is based on peak detection of the spectrum magnitude at particular frequencies for obtaining the quantitative expressions. The radiofrequency signals of ULWs (ultrasonic Lamb waves) were used for the accuracy evaluation of the method. An uncertainty evaluation was conducted to guarantee the metrological traceability of measurement results and ensure that they are accurate and reliable. Mathematical and experimental verifications were conducted by using signals of Lamb waves propagating in the aluminum plate. The obtained mean relative error of 0.12% for the A_0_ mode (160 kHz) and 0.05% for the S_0_ mode (700 kHz) during the mathematical verification indicated that the proposed method is particularly suitable for evaluating the phase-velocity dispersion in clearly expressed dispersion zones. The uncertainty analysis showed that the plate thickness, the mathematical modeling, and the step of the scanner have a significant impact on the estimated uncertainty of the phase velocity for the A_0_ mode. Those components of uncertainty prevail and make about ~92% of the total standard uncertainty in a clearly expressed dispersion range. The S_0_ mode analysis in the non-dispersion zone indicates that the repeatability of velocity variations, fluctuations of the frequency of Lamb waves, and the scanning step of the scanner influence significantly the combined uncertainty and represent 98% of the total uncertainty.

## 1. Introduction

In 1991, Alleyne and Cawley were the first [1] to propose the use of the two-dimensional fast Fourier transform (2D-FFT) for measuring propagating multimode signals. Since then, the 2D-FFT has been widely applied and used in the field of signal processing and, therefore, described as a classical and/or traditional method. The basic premise of this method is to transform the received amplitude–time record into amplitude–wavenumber records at discrete frequencies [1,2]. Using the 2D-FFT technique, the wave propagating along the object during the experiment is characterized by the distance (*d*) and time (*t*) that are transformed into the wavenumber (*k*) and frequency (*f*) space. As a result, the image (*t*,*f*) of the measured array signals is projected [3,4]. By applying special signal-processing algorithms, the time and frequency (*t*,*f*) domain can be determined. It is the primary condition to analyze the non-stationary/multimode signals, such as ultrasonic Lamb waves (ULWs).

Ultrasonic inspection is becoming a standard method in Non-Destructive Testing (NDT) and Structural Health Monitoring (SHM) applications [5]. As a result, the signals of ULWs are among many other non-stationary/multimode signals that are widely used in civil, mechanical, and aerospace industries for detecting internal structural defects, their location and sizing, structural discontinuities, material parameters, and others [6]. However, the ULWs possess the dispersion phenomenon (the velocity of waves varies depending on the frequency and thickness); that is, the signal-processing methods should be able to perform the calculation of both (*t*,*f*) domains and allow displaying them [7,8,9]. Another undesirable phenomenon is an infinite number of modes. Lamb waves have an infinite number of the dispersive symmetric (S_n_) and antisymmetric (A_n_) modes, which emerge depending on the thickness of the object, the material under investigation, and the frequency; and every one of them is described by two velocities, namely phase and group [9,10]. Therefore, special signal-processing algorithms need to be applied to examine the signals of Lamb waves, except for the analysis of the evanescent modes. That is why the 2D-FFT method, which enables us to identify the multimodal dispersion modes and display them from the corresponding (*f*,*k*) energy trajectories, is considered to be a good tool for the analysis of such a signal. However, there is one significant drawback to this method. The obtained results are presented in images, and additional algorithms have to be used for the quantitative expressions. As a result, a number of various signal-processing algorithms are created and used for the quantitative display of (*f*,*k*) domains for the dispersion evaluation of Lamb waves [10,11]. Then the representation can be directly associated with the calculation of the velocity changes, which are used for detecting defects and/or delamination [12] and provide understanding about the wave-propagation phenomena. Moreover, wave propagation and interaction effects, such as reflections, refractions, diffractions, mode conversions, and others, influence the analysis of phase-velocity dispersion due to distortions of waveforms [5]. As presented in Reference [12], not only the location of the defect but also its size is determined according to the changes of the phase velocity. Therefore, algorithms allowing direct reconstruction of the quantitative values of the dispersion curves by (*f*,*v*) are constantly developed and researched. However, it should be noted that, in order to estimate the location and size of the defect based on phase-velocity changes, at first it is necessary to know the reliability of these methods. Therefore, an assessment of the accuracy of any method developed must be performed.

In the presented work, the focus is on the reliability evaluation of the 2D-FFT method that is based on peak detection spectrum magnitude at particular frequencies. This signal-processing method was developed and presented in our previous work [13]. The investigation was carried out by using Lamb wave signals propagating in different materials and geometry objects. The obtained results showed that the method is appropriate for evaluating the phase-velocity dispersion of the Lamb waves.

Validation of the signal-processing method is an important and necessary part, as it helps to avoid costly and time-consuming practices and ensure the comprehension of the method efficiency [14]. All investigations should be performed to quantify an optimal uncertainty, as to avoid new testing. Using the obtained results, the analysis of complex parameters can be minimized, thus minimizing costs and demonstrating that further testing or calculations will not enhance the quality of the predictive/expected measurement. It is an important aspect of measurement that affects costs, quality, decisions, and risks of decisions taken. According to the presented work [14], such a procedure should include several main steps:(1)Development of an algorithm of the signal-processing method.(2)Mathematical modeling of the dispersion curve segments. The purpose is to identify the capabilities of the most accurate reconstructed segments and determine the uncertainty components associated with the model errors.(3)Experimental setup and verification of the results.(4)Determination of input and output parameters, which affect the final method accuracy. Optimization of the selected uncertainty components for uncertainty quantification in high-dispersion and non-dispersion zones of phase-velocity dispersion curves of the A_0_ and S_0_ modes.

The main task of the presented work was to evaluate the accuracy of the 2D-FFT method based on the peak detection of the spectrum magnitude at certain frequencies, using the signals of Lamb waves propagating in a homogeneous aluminum plate. Using the general principles of measurement theory, we identify, analyze, and present the main characteristics and sources of uncertainties that mainly affect the accuracy of the obtained results. It benefits laboratories that evaluate the uncertainty of their measurement results when demonstrating their technical competence upon their certification/audit in accordance with SO/IEC 17025:2017.

The paper is organized as follows: The technique of peak detection of the spectrum magnitude from the 2D-FFT image is introduced in Section 2. A mathematical simulation of the object and numerical verification of the 2D-FFT method are presented in Section 3. The experimental verification of the proposed method is described in Section 4. An analysis of the uncertainties and limitations is discussed in Section 5. The conclusions of the research are presented in Section 6.

## 2. The Technique of Peak Detection of the Spectrum Magnitude

As mentioned in the introduction, using the 2D-FFT method, the distance and time are transformed into the wavenumber and frequency space, and an image of a 2D data array is obtained. To obtain quantitative expressions, the technique of peak detection of 2D spectrum magnitude is proposed, which was presented in the previous work [14]. Therefore, at this stage, the general flowchart of the proposed algorithm with the main steps (Figure 1) and short explanations is presented below.

The algorithm consists of two main stages: the 2D-FFT method application and calculation of peak detection of 2D spectrum magnitude of the acquired signals:I.Application of the 2D-FFT method for analysis of the B-scan data. Since the 2D-FFT method is well-known, the mathematical equations are not given, but the segment of the A_0_ mode phase-velocity curve obtained by this method is displayed in Figure 1.II.Detection of 2D spectrum magnitude peaks includes the following steps:Selecting the frequency bandwidth (from *f*_1_ up to *f*_2_).Estimating the phase velocity of Lamb wave modes from the 2D-FFT image and applying the peak detection of 2D spectrum magnitude at maximum energy and particular frequencies (within the selected frequency bandwidth).The second-order polynomial approximation is applied in order to reduce the influence of scattering effects of detected peaks of phase velocity due to the presence of blurred shapes of 2D spectrum magnitude.


## 3. The Object Mathematical Simulation and Numerical Verification of the 2D-FFT Method

### 3.1. The Object Mathematical Simulation

Based on the methodology for the reliability evaluation of the signal-processing methods used for the dispersion estimation of Lamb waves presented by Draudviliene and Meskuotiene [13], the mathematical verification should be conducted at the beginning. Since experimental research is necessary for the complete assessment of the method reliability, the properties and geometry of the chosen object should be the same in the mathematical modeling and experimental research. Thus, the study is conducted by using a homogeneous 7075-T6 aluminum plate [15]. The material properties of such a 2 mm–thick aluminum plate are as follows: density, *ρ* = 2780 kg/m^3^; Young’s modulus, *E* = 71.78 GPa; and Poisson’s ratio, *ν* = 0.3435.

The next step of the presented study is a selection of the frequency range. For that purpose, the phase-velocity dispersion curves of the asymmetric and symmetric modes of Lamb waves are required. The analytical computational package ‘DISPERSE’ was chosen [16], and using the geometry parameters and material properties of the selected object, we plotted the calculated dispersion curves, which are shown in Figure 2.

Based on the obtained phase-velocity dispersion curves, the frequency range up to 300 kHz and possessing a central frequency of 160 kHz is selected for the study, where the A_0_ mode is highly dispersive; meanwhile, the S_0_ mode shows a weak dispersion level. Thus, the three-period harmonic bursts with Gaussian envelop that have a frequency of 160 kHz are used as the input signal, $u_{0}\left(t\right)$. The propagating signals are then obtained according to the following [17]:(1)$$u_{x}\left(t\right)=IFT[FT[u_{0}\left(t\right)]\cdot H\left(jf,x\right)]$$
where $u_{x}\left(t\right)$ is the output signal, and $u_{0}\left(t\right)$ is the input signal. The *IFT* denotes the inverse Fourier transform; H(jf,x) is the complex transfer function of the object given by $H\left(jf,x\right)=e^{-\alpha \left(f\right)x}e^{-j\omega \frac{x}{c_{ph}\left(f\right)}}$; *x* is the propagation distance, α(f) is the frequency-dependent attenuation coefficient; $c_{ph}\left(f\right)$ is the phase-velocity dispersion curve corresponding to the particular guided wave mode; *ω* is the angular frequency; and *j* is the basic imaginary unit, $j=\sqrt{-1}$. Since the attenuation of the Lamb waves is very low in the case of the unloaded metal plates, this parameter is eliminated.

A 200 mm distance with a d*x* step of 0.1 mm is used to obtain the signals of both A_0_ and S_0_ modes. In this way, 2001 simulated signals for both modes are obtained, which are displayed in two B-scan images. The images of B-scans of the A_0_ and S_0_ modes are presented in Figure 3a,b, respectively. The normalized amplitudes of the signals are presented by the appropriate color indicated in color bar.

Having the simulated signals, the 2D-FFT method and technique of peak detection spectrum magnitude can then be applied. Likewise, the technique reliability evaluation at the theoretical level can be obtained.

### 3.2. Mathematical Verification of the Method

#### 3.2.1. Investigation of the A_0_ Mode

Using the 2D-FFT method, the A_0_-mode phase-velocity dispersion curve segment at the 160 kHz frequency range was reconstructed, and the image is presented in Figure 4a. Then, using the presented technique of detection of peaks of the spectrum magnitude, the phase-velocity segments of the A_0_ mode in numerical values were retrieved. The reconstructed A_0_-mode phase-velocity segment, which was created by calculating the peak values of spectrum magnitude at particular frequencies, is presented in Figure 4b. The reconstructed segment covers the (114–214 kHz) frequency range. Then, using the analytical method (computational package ‘DISPERSE’) as a reference method, the comparison of the obtained results was conducted, and the obtained results are presented in Figure 4c.

According to Reference [13], the final step at the theoretical level is the calculation of the mean values of absolute and relative errors and standard deviation by comparing the results obtained by the mathematical simulation and reference method.

The absolute error and the average of absolute errors are calculated as follows:(2)$$\Delta_{c_{mat,\;n}}=c_{mat}\left(f_{n}\right)-c_{ref}\left(f_{n}\right)$$
(3)$${\bar{\Delta }}_{c_{mat}}=\frac{1}{N}\overset{N}{\underset{n=1}{\sum }}\Delta_{c_{mat,\;n}},$$
where *n* = 1, …, *N, n*th point of the dispersion curve, *N* is the number of points in a segment of the dispersion curves, $c_{ref}\left(f_{n}\right)$ is the phase velocity at the corresponding frequency obtained according to the reference dispersion curve, and $c_{mat}\left(f_{n}\right)$ is the phase velocity at the corresponding frequency of the reconstructed dispersion curve obtained by the mathematical simulation. Then the standard deviation is calculated according to the following:(4)$$\sigma_{\Delta_{c_{mat}}}=\sqrt{\frac{\sum_{n=1}^{N}{\left(\left(\Delta_{c}{}_{mat,n}\right)-{\bar{\Delta }}_{c_{mat}}\right)}^{2}}{\left(N-1\right)}}$$

The standard deviation represents the reliability of the method at the theoretical level, and it will be included in the uncertainty budget. The obtained results are presented in Table 1.

The phase-velocity dispersion curve was reconstructed in the frequency domain covering 160 kHz, with an average relative error of 0.12%. Based on the result, one can assume that the proposed method is a reliable tool for the evaluation of the A_0_-mode phase-velocity dispersion and reconstruction dispersion curve segment in a clearly expressed dispersion zone. The repeatability of the variation of the measured points of dispersion curves is a part of the combined standard uncertainty and is equal to 0.2%. The relative standard deviation was calculated as follows: $\sigma_{\Delta_{c_{mat}}}\times 100\%/1590$.

#### 3.2.2. Examining the S_0_ Mode

The same procedure was applied for analyzing the suitability of the proposed technique of the peak detection spectrum magnitude for evaluating the phase-velocity dispersion by reconstructing the dispersion curve of S_0_ mode. The retrieved results are presented in Figure 5.

The comparison of the obtained results (Figure 5c) shows the significant deviation from the reference segment of the dispersion curve of the phase velocity. It should be noted that, in the case of the A_0_ mode, which is analyzed in the very high dispersion zone, the different frequency components propagate at different velocities and are distributed in a wide frequency range, which covers (114–214 kHz; see Figure 4c). Thus, the detection of the peak spectrum magnitudes at the particular frequencies and the conduct of the second-order polynomial approximation are the appropriate solutions. Meanwhile, in the case of the S_0_ mode, the research was conducted in a non-dispersive zone. Due to the concentration of the different frequency components in a relatively narrow frequency range, from 140 kHz up to 180 kHz (Figure 5a), the detected values of the peak spectrum magnitudes in the frequency axis and velocity axis are slightly distributed within the circularly shaped region. As the detected peak values coincide, the reconstructed phase-velocity dispersion curve has only a few values (Figure 5c); thus, estimating their relation to the set of particular phase-velocity and frequency values is complicated enough. The studies carried out [18] on the S_0_ mode in a non-dispersive frequency range (central frequency of 300 kHz) showed that variations of the phase-velocity components are covering a very narrow range of distribution. Thus, the second-order polynomial approximation of more densely concentrated phase-velocity values in terms of frequency generates some errors. The obtained discrepancy of values is presented in Figure 5c.

Thus, in order to summarize the efficiency of the proposed method for using the dispersion evaluation of Lamb waves, an additional study needs to be conducted. In relation to that, the frequency range where the S_0_ mode possesses a high dispersion nature is selected.

As Figure 6 demonstrates, the S_0_ mode at the central frequency of 700 kHz (frequency range from 400 kHz up to 1000 kHz) possesses a very high dispersion nature. Thus, the three-period harmonic bursts with the Gaussian envelope and having a frequency of 700 kHz are used for simulating the input signal $u_{0}\left(t\right)$ in order to obtain the B-scan image of the S_0_ mode (Figure 6). All other parameters are the same as those that were previously used for the research at the central frequency of 160 kHz.

Having the simulated signals and B-scan images, the 2D-FFT method and the technique of peak detection spectrum magnitude were applied, and the retrieved results are presented in Figure 7a,b respectively. The obtained segment comparison of the phase-velocity dispersion curve with the analytical method that uses the computational package ‘DISPERSE’ is presented in Figure 7c.

The comparison of the obtained results (Figure 7c) shows the high accuracy of the method when reproducing dispersion curves in clearly expressed dispersion ranges.

Using Equations (2)–(4), we performed the calculations of the mean absolute, relative errors, and standard deviation, and these are presented in Table 2.

The mean relative error for the S_0_ mode (Table 2) calculated in the region of the significant phase-velocity dispersion (700 kHz) is ten times smaller than the obtained one in the non-dispersive zone (160 kHz).

In order to complete the assessment of the reliability of the proposed method, experimental research was conducted.

## 4. Experimental Verification

### 4.1. Description of the Experimental Setup

An isotropic aluminum plate (dimensions 1.2 × 1.2 m^2^) of 2 mm in thickness was chosen for the experimental research of the Lamb-wave propagation properties. The parameters of the aluminum alloy plate used in the experiments were the same as those in the mathematical verification and described in Section 3.1. The structural scheme of the experimental equipment used in the study and arrangement of the ultrasonic transducers are presented in Figure 8.

The experimental study was performed by using low-frequency, wideband contact-type ultrasonic transducers with 180 kHz resonant frequency developed at the Ultrasound Research Institute. The frequency bandwidth of these transducers is from 40 kHz up to 640 kHz (at −10 dB). A detailed description of the transducers is provided in Reference [19]. The two ultrasonic transducers are used to excite and receive the Lamb waves; one of transducer (the transmitter) is mounted on a selected location on the plate, and the other (the receiver) is repositioned by the linear mechanical scanner. The diameter of the active size of the spherically shaped sensor surface is 1 mm. Thus, a point-like transmitter–receiver effect is achieved, with spatial dimensions significantly smaller than the wavelengths of the generated modes. The resonant frequency of the contact transducers is 160 kHz, and the transmitter is excited by a 3-period Gaussian envelope signal. The position of the receiver is changed with a linear scanner Standa 8MTF-75LS05 (Standa Ltd., Vilnius, Lithuania). The experiments were conducted by using the ultrasonic measurement system ULTRALAB. This system was designed and developed at the Ultrasound Institute of the Kaunas University of Technology and consists of a voltage generator, a low-noise amplifier, and an analogue-to-digital converter (ADC). The sampling frequency of the ADC is 50 MHz. A low-noise 13.4 dB preamplifier was connected to the ultrasonic receiver to increase the signal level and improve the signal-to-noise ratio. In order to form the B-scan image, the transmitter was attached at a fixed position to the top surface of the aluminum plate. The receiver was scanned in the distance range of 178 mm up to 278 mm away from the excitation point (transmitter) with the d*x* = 0.1 mm scanning step. The formed B-scan image is displayed in Figure 9a. This image clearly shows two different mode signals. The moving time windows are used to distinguish these modes. The dotted lines in Figure 9a indicate the limits of these windows. The distinguished S_0_ and A_0_ modes are normalized according to the maximum amplitudes and are displayed in Figure 9b and Figure 9c, respectively.

### 4.2. Experimental Verification of the Analyzed Method

The same procedure as in the mathematical investigation is used to analyze the reliability of the method for the experimentally obtained signals. The images of the phase-velocity dispersion curve segments of the A_0_ and S_0_ modes from using the 2D-FFT method and the numerical estimations of phase-velocity values from using the technique of peak detection of spectrum magnitude were obtained and are presented in Figure 10a,b, respectively. Then the analytical method (using the computational package ’DISPERSE’) was applied as a reference method and the comparison of the obtained results was conducted and is presented in Figure 10c. The analysis was performed by using the experimentally retrieved signals propagating along with the plate whose material parameters (Young’s modulus, Poisson’s ratio, and density) and geometry were used in the mathematical verification. The study was conducted in the same frequency range for different modes (the A_0_ and S_0_). The results obtained in each case were compared by using the same reference method.

As mentioned above, the S_0_ mode is analyzed in the non-dispersive range; therefore, the detected peak values coincide, and the dispersion curve is reconstructed in a narrower frequency range.

In order to evaluate the quantitative characteristics of the method accuracy, a systematic error of the experimental values from conventional true values should be estimated [20,21]. The systematic error of velocity is performed by comparing the segments of the dispersion curves reconstructed in both ways in the same frequency range. Then the systematic error and the average of the errors are obtained as follows:(5)$$\Delta_{c_{k}}=c_{ex}\left(f_{k}\right)-c_{ref}\left(f_{k}\right),$$
(6)$${\bar{\Delta }}_{c}=\frac{1}{K}\overset{K}{\underset{k=1}{\sum }}\Delta_{c_{k}},$$
where *k* = 1,…; *K* is the *k*th point of the segment; *K* is the number of points in a segment of the dispersion curves; $c_{ex}\left(f_{k}\right)$ is the phase velocity at the corresponding frequency of the reconstructed dispersion curve from the experimental signals; and $c_{ref}\left(f_{k}\right)$ is the phase velocity at the corresponding frequency obtained according to the reference dispersion curve.

The scatter of values obtained from repeated measurements is characterized by the standard deviation of phase velocity, which is calculated as follows:(7)$$\sigma_{\Delta_{c}}=\sqrt{\frac{\sum_{k=1}^{K}{\left(\Delta_{c}{}_{k}-{\bar{\Delta }}_{c}\right)}^{2}}{\left(K-1\right)}}.$$

The mean absolute and relative errors for the A_0_ and S_0_ modes and the standard deviation are presented in Table 3. The obtained results were used in the uncertainty analysis.

By analyzing the data presented in Table 1, the negative mean systematic error of the S_0_ mode was obtained. As values of the reconstructed dispersion curve from the experimental signals at the corresponding points for the S_0_ mode were lower than the corresponding values of the reference dispersion curve (Figure 10c), a negative mean systematic error for the S_0_ mode was obtained.

## 5. Analysis of Uncertainties

The uncertainty calculation methodology defined in our previous work [13] was applied for the uncertainty calculation. Uncertainties were calculated for a homogeneous material whose density and elastic constants were specified in the previous section. A complete uncertainty budget was determined for each mode. The cases under consideration include the clearly expressed dispersion and ‘non-dispersion’ zones of the reconstructed curves.

To evaluate the influence of extraneous factors or errors on the uncertainty of the measurement result, the function of velocity variation was investigated. The components of uncertainty discussed above, such as the standard deviation ($\sigma_{\Delta_{c_{mod}}}$) of velocity errors reflecting the influence of the mathematical model and the standard deviation ($\;\sigma_{\Delta_{c}}$) of velocity errors obtained from repeated experimental measurements, are included in the combined standard uncertainty. In order to estimate fluctuations of frequency of the Lamb waves and their influence on the phase velocity, which is directly affected by the receiver characteristics [13], the maximum deviation from the average error for one point $\sigma \left(\Delta_{c_{phmax}}\right)\;$ was calculated. To estimate the limits of actual variability of the obtained results, the difference between the limit value of the absolute error and the mean absolute error was used. The statistical distribution associated with the input sources, namely $\sigma_{\Delta_{c_{ph}}}$, $\sigma_{\Delta_{c_{mod}}},\;\mathrm{and}\;\;u_{\Delta_{c_{phmax}}}$, is considered to be normal or Gaussian. The difference between two neighboring points at which signals were received was *l =* 0.1 mm. The distance, *l*, was determined with a standard uncertainty of $\left(\Delta_{l}\right)=\pm \frac{\Delta l}{\sqrt{3}}$, where Δl is the error of step setting equal to 0.05 mm. The sensitivity coefficient, *W*_Δ*l*_, is equal to 1/*t*, where *t* is a rectangular single pulse duration of 1.67 μs. The uniform distribution is used for sources related to the test object parameters because the range of values is known, that is, an interval with the minimum and maximum values. The sensitivity coefficient of the measurement in relation to the test object parameters was determined experimentally. The dispersion curves of the Lamb waves are non-linear functions of the test object parameters (density, *ρ*; Young’s modulus, *E*; the Poisson’s ratio, *υ*; the plate thickness, *d*; and distance between two points, *l**)* and the ultrasonic signal frequency, *f*. Likewise, the approximate Δ*F(x_i_)* is evaluated with a small deflection, Δ*x_i_*, of the variable *x_i_*:(8)$$\Delta F\left(x_{i}\right)=\frac{\left[F\left(x_{i}\right)-F\left(x_{i}-\Delta x_{i}\right)\right]}{2}+\frac{\left[F\left(x_{i}+\Delta x_{i}\right)-F\left(x_{i}\right)\right]}{2}\;.$$

The sensitivity coefficients can be evaluated as follows:(9)$$W_{i}\left(x_{i}\right)=\frac{\Delta F\left(x_{i}\right)}{\Delta x_{i}},$$
where $\Delta x_{i}$ is the small change of the variable $x_{i}$, and $\Delta F\left(x_{i}\right)$ is the function change due to the change of the variable $x_{i}$. The input data used to estimate the sensitivity coefficient of the test object parameters are presented in Table 4. The reference phase velocity is 1590 m/s for the A_0_ mode and 5402 m/s for the S_0_ mode at the central frequency, *f* = 160 kHz. The change of the parameter Δ*x_i_* is equal to 20% of the actual value.

The components of the combined uncertainty were processed and analyzed by using GUM Workbench version 2.4.1.384 software (Metrodata GmbH, Braunschweig, Germany). The GUM uncertainty framework is based on the law of propagation of uncertainties (LPUs) [21]. The combined uncertainty of each reconstructed frequency range was also calculated by using the Monte Carlo simulation. We determined that this methodology carries more information than the simple propagation of uncertainties and generally provides results that are closer to reality [22]. This mathematical technique is integrated into the GUM Workbench version 2.4.1.384 software. The input values used for data processing according to GUM methodology and Monte Carlo simulation are described in Table 5 and Table 6 and Figure 11. The characteristics evaluated by both methods are comparable. Therefore, we consider the results obtained to be reliable. All results are displayed in Table 5 and Table 6 and Figure 11. Implementation of the 2D-FFT is more accurate in the clearly expressed dispersion zone. The result of the velocity measurement for the A_0_ mode in the range (114.3–213.6) kHz is (1590 m/s + 0.6%) ± 2.6%. The result of the velocity measurement for the S_0_ mode in the range (137.5–186.5) kHz is (5402 m/s − 1.6%) ± 1.6%. These results are presented in the following form: (*c*$\;\pm \;{\bar{\delta }}_{c}$) $\pm \;U\left(\delta_{c}\right)$*,* where *c* is the measured value of velocity, ${\bar{\delta }}_{c}\;$ is the mean relative error, and the range ±$U\left(\delta_{c}\right)$ is the expanded uncertainty.

As observed in Table 5 and Table 6, the dominant influence on the overall uncertainty is attributed to the plate thickness, mathematical modeling, and step of the scanner, which is directly related to the distance between two points. The overall uncertainty in the ‘non-dispersion’ zone particularly depends on several factors, including the uncertainty attributed to the repetition of velocity variations, fluctuations of the Lamb wave’s frequency, and the scanning step of the mechanical scanner.

We performed a comparison between the obtained results for our method and the results obtained by using other phase velocity evaluation methods. Using the zero-crossing technique, we determined that the excitation frequency for both modes was 300 kHz, which means that the S_0_ mode was studied in the non-dispersive zone; meanwhile the A_0_ was in the clearly expressed dispersion zone [18]. Thus, the comparison of the obtained systematic errors shows that the 2D-FFT technique lets us reconstruct the phase-velocity dispersion curves for the A_0_ mode more accurately, only with a 0.6% mean relative error. Meanwhile, the phase velocity of the S_0_ mode was reconstructed with a 0.65% mean systematic error, using the zero-crossing technique. Moreover, it is 2.5 times more accurate than in the case of the 2D-FFT method. The assessment of the reliability of developed methods indicates that the 2D-FFT method is more suitable to be used in the studies, which provide an evaluation of the clearly express dispersive zones. Thus, assessing the reliability of developed methods is a necessary task that can be used to assess the accuracy of developed methods compared to other methods and to make it easier to choose the most appropriate method needed to solve the specific problem or task.

## 6. Conclusions

An implementation of the 2D-FFT method based on the technique of peak detection of the spectrum magnitude at particular frequencies was presented. Mathematical and experimental verifications using signals of Lamb waves propagating in the aluminum plate were performed. A major focus was given to the experimental results obtained for the systematic error, which plays a significant role in the overall error contribution and expanded uncertainty, which characterizes the quality of measurement results. The obtained mean systematic error of 0.6% for the A_0_ mode at 160 kHz and 0.05% for the S_0_ mode at 700 kHz in the mathematical investigation indicate that the proposed method is particularly suitable for evaluating the phase-velocity dispersion by reconstructing dispersion curves in clearly expressed dispersion zones. The uncertainty analysis has shown the components that significantly affect the measurement result under different conditions. The plate thickness, mathematical modeling, and scanning step of the mechanical scanner have a significant impact on the estimated uncertainty of the phase velocity for the A_0_ mode. Those components dominate and make about ~92% of the total standard uncertainty. The combined uncertainty in the non-dispersive zone for the S_0_ mode is sensitive to the repeatability of velocity variations, fluctuations of the Lamb wave’s frequency, and the scanning step of the scanner. These three components represent 98% of the total uncertainty. The obtained results for the phase-velocity measurement for the A_0_ mode in the range (114.3–213.6) kHz and the S_0_ mode in the range (137.5–186.5) indicate that the proposed method is a reliable tool for the quantitative evaluation of the phase velocity using Lamb waves. This will facilitate the selection of the method according to the required parameters, solving one or another problem in various future studies.

## Figures and Tables

**Figure 1 sensors-22-06750-f001:**
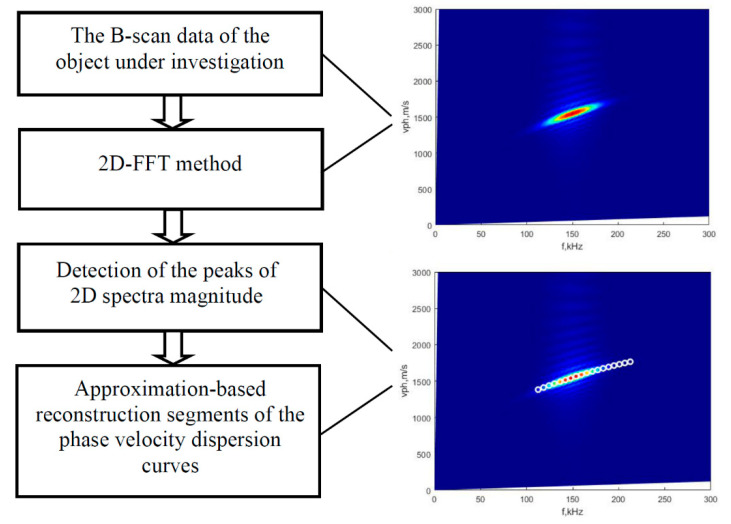
An algorithm of the 2D-FFT method based on peak detection of 2D spectrum magnitude.

**Figure 2 sensors-22-06750-f002:**
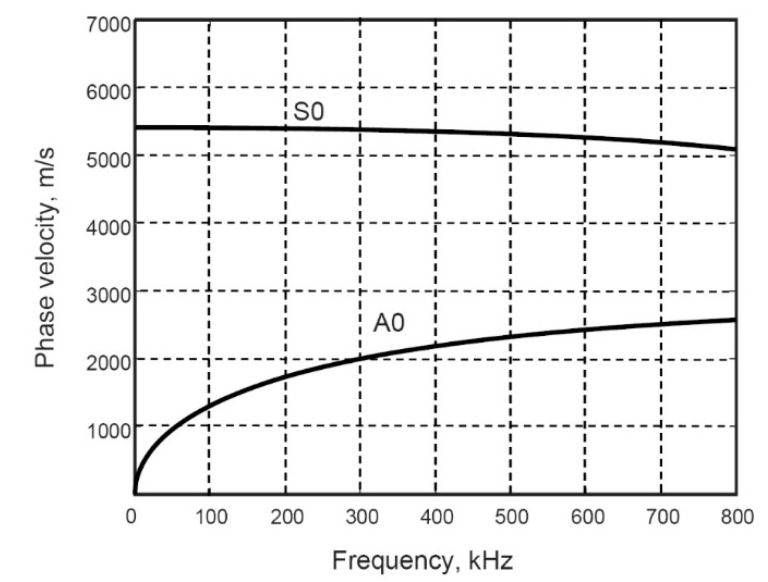
The phase-velocity dispersive curves of the A_0_ and S_0_ modes of Lamb waves calculated by using the analytical method (package ‘DISPERSE’).

**Figure 3 sensors-22-06750-f003:**
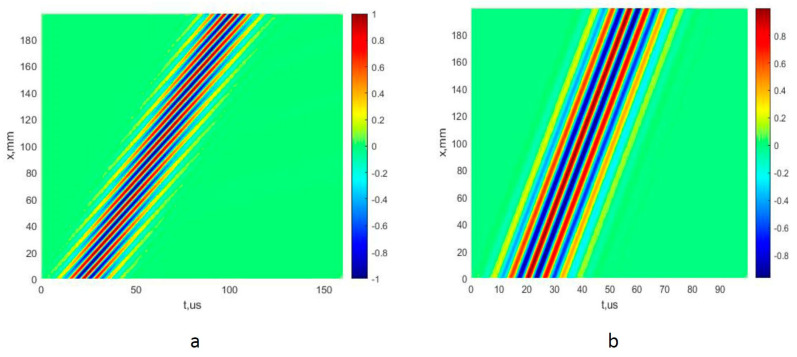
The B-scan images of the simulated A_0_ (**a**) and S_0_ (**b**) modes of Lamb waves, using a mathematical model.

**Figure 4 sensors-22-06750-f004:**
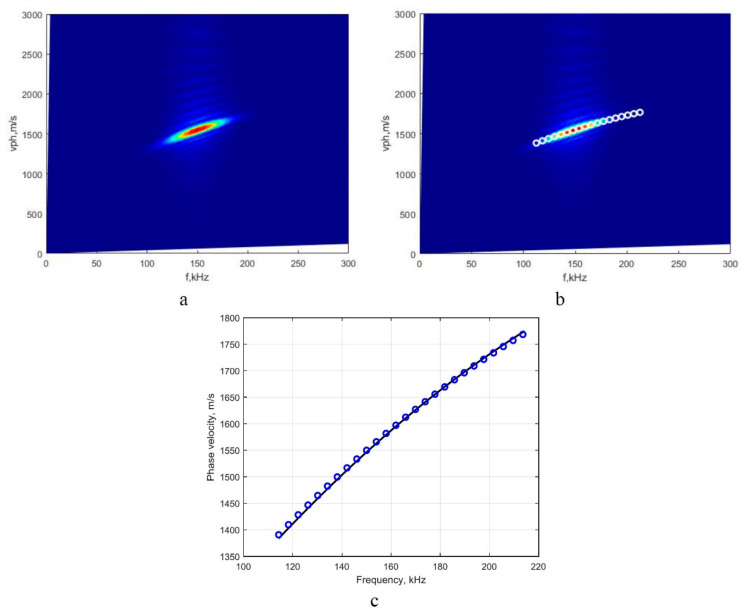
Image of the A_0_-mode phase-velocity dispersion curve segment obtained by 2D-FFT method (**a**); reconstructed quantitative values of the dispersion curves segment, using the proposed method (**b**); and the dispersion curve segment obtained by analytical method (package ‘DISPERSE’, the solid line) and reconstructed by using the proposed algorithm (circles) (**c**).

**Figure 5 sensors-22-06750-f005:**
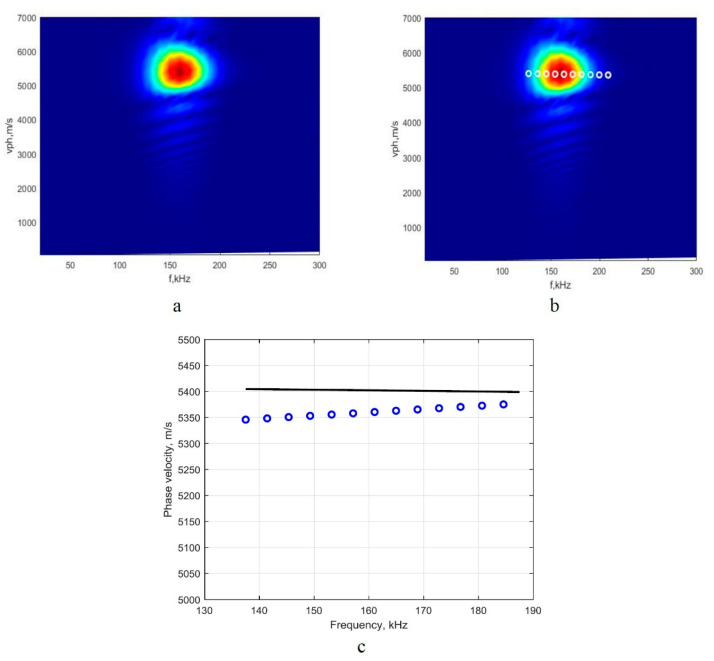
Image of the S_0_-mode phase-velocity dispersion curve segment obtained by 2D-FFT method (**a**); reconstructed quantitative values of the dispersion curves segment, using the analyzed method (**b**); and the dispersion curve segment obtained by using the analytical method (package ’DISPERSE’, the solid line) and reconstructed by using the analyzed algorithm (circles) (**c**).

**Figure 6 sensors-22-06750-f006:**
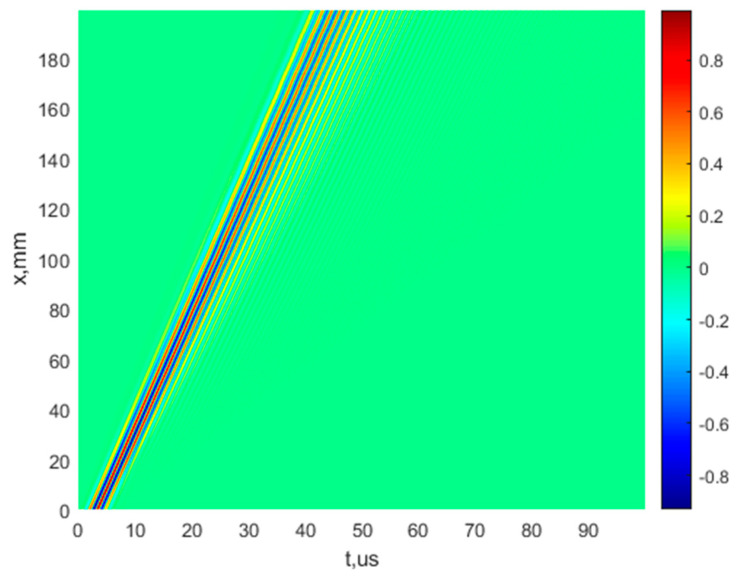
Simulated B-scan image of the S_0_ mode at the central frequency of 700 kHz, using the mathematical model.

**Figure 7 sensors-22-06750-f007:**
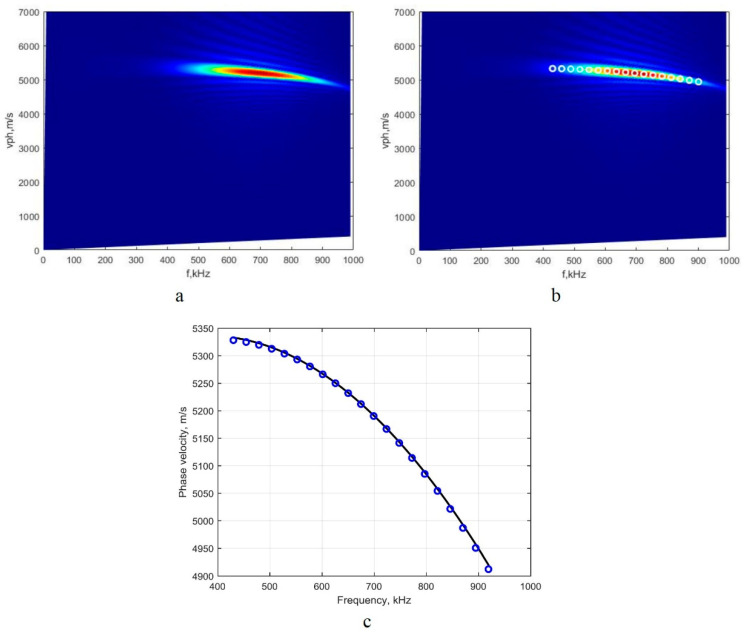
The image of the S_0_-mode phase-velocity dispersion curve segment obtained in the 700 kHz range by the 2D-FFT method (**a**), the quantitative values of the dispersion curve segment that were reconstructed by using the analyzed method (**b**), and the dispersion curve segment acquired by using the analytical method (the solid line) and reconstructed by using the proposed algorithm (circles) (**c**).

**Figure 8 sensors-22-06750-f008:**
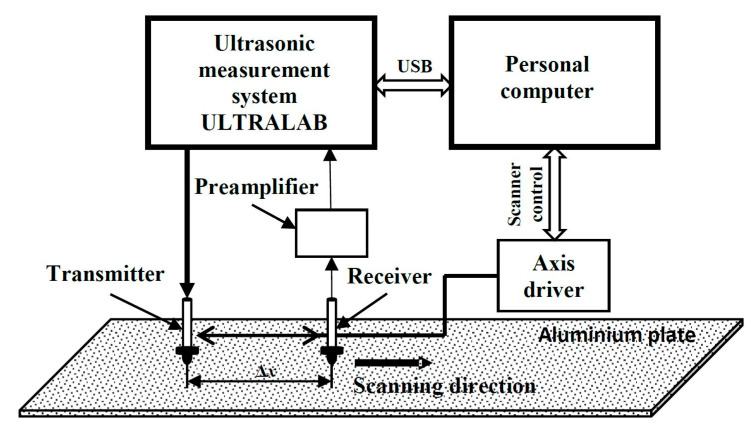
The experimental setup for generating and receiving Lamb wave signals on an aluminum plate.

**Figure 9 sensors-22-06750-f009:**
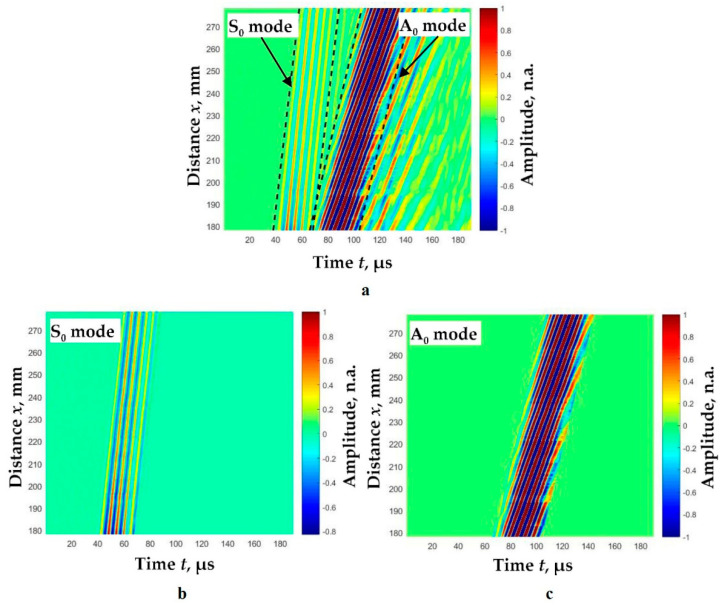
Experimental B-scan image of the Lamb wave A_0_ and S_0_ modes propagating along the aluminum plate of 2 mm in thickness (**a**) and S_0_ (**b**) and A_0_ (**c**) modes separated by a moving time window (**c**). The abbreviation ‘n.a.’ means normalized amplitude units.

**Figure 10 sensors-22-06750-f010:**
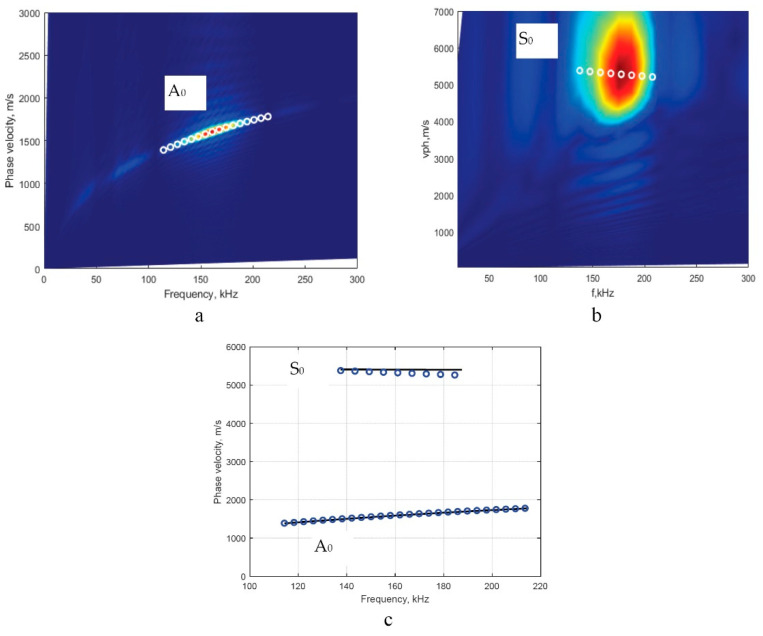
The phase-velocity dispersion curves are calculated by 2D-FFT and peak detection of the spectrum magnitude of 2D FFT (circles) for the A_0_ (**a**) and S_0_ (**b**) modes; and the dispersion curve segment obtained by using the analytical method (by computational package ‘DISPERSE’, the solid line) and reconstructed by using the proposed algorithm (circles) (**c**).

**Figure 11 sensors-22-06750-f011:**
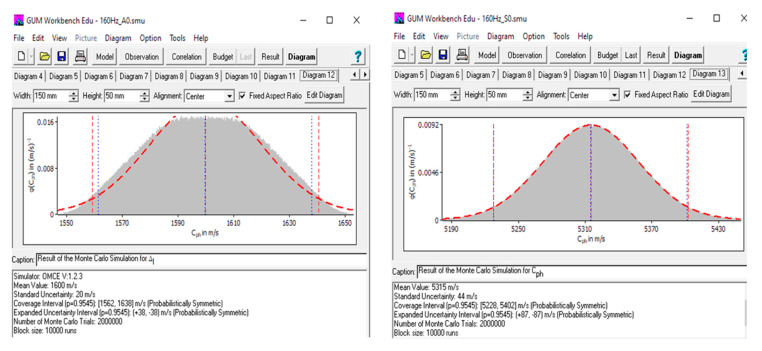
Expanded uncertainty of the reconstructed frequency range for the A_0_ and S_0_ modes, respectively, calculated by using Monte Carlo simulations.

**Table 1 sensors-22-06750-t001:** The metrological characteristics of the phase-velocity (*v* = 1590 m/s) dispersion curve for the A_0_ mode.

$\mathrm{Mean}\;\mathrm{Absolute}\;\mathrm{Error}\;{\bar{\Delta }}_{c_{mat}},\;m/s$	$\mathrm{Standard}\;\mathrm{Deviation}\;\sigma_{\Delta_{c_{mat}}},m/s$	$\mathrm{Mean}\;\mathrm{Relative}\;\mathrm{Error}\;{\bar{\delta }}_{c_{mat}},\;\%$
1.68	3.28	0.12

**Table 2 sensors-22-06750-t002:** The metrological characteristics of the phase-velocity (*v* = 5402 m/s) dispersion curve for the S_0_ mode.

Frequency, kHz	$\mathrm{Mean}\;\mathrm{Absolute}\;\mathrm{Error}\;{\bar{\Delta }}_{c_{mod}},\;m/s$	$\mathrm{Standard}\;\mathrm{Deviation}\;\sigma_{\Delta_{c_{mod}}},\;m/s$	$\mathrm{Mean}\;\mathrm{Relative}\;\mathrm{Error}\;{\bar{\delta }}_{c_{mod}},\;\%$
160	29.1	5.79	0.54
700	2.61	1.87	0.05

**Table 3 sensors-22-06750-t003:** The metrological characteristics of the velocity (*f* = 160 kHz) dispersion curve for the A_0_ and S_0_ modes.

Velocity, m/s	Mean Absolute Error ${\bar{\Delta }}_{c},\;m/s$	$\mathrm{Standard}\;\mathrm{Deviation}\;\sigma_{\Delta_{c}},m/s$	Mean Relative Error ${\bar{\delta }}_{c},\;\%$
A_0_ mode			
1590	9.85	1.67	0.62
S_0_ mode			
5315	−86.84	34.77	1.61

**Table 4 sensors-22-06750-t004:** Phase-velocity dependence on the material density, *ρ*; Young’s modulus, *E*; the Poisson’s ratio, *υ*; and the plate thickness, *d*, for the A_0_ and S_0_ modes.

Object Parameter *x_i_*	Parameter Change Δ*x_i_*	Velocity Change Δ*Fx_i_*, m/s	Sensitivity Coefficient *Wx_i_*
*ρ* = 2780 kg/m^3^	Δ*ρ* = 556 kg/m^3^	A_0_ mode	
		110	$0.2\;m^{4}/s\cdot \mathrm{kg}$
		S_0_ mode	
		640	$1.2\;m^{4}/s\cdot \mathrm{kg}$
*υ =* 0.3435	Δ*υ* *=* 0.0687	A_0_ mode	
		14	200 m/s
		S_0_ mode	
		122	170 m/s
*E* = 71.787 GPa	Δ*E* = 14.357 GPa	A_0_ mode	
		105	7.3 m/s·GPa
		S_0_ mode	
		572	40 m/s·GPa
*d* = 2 mm	Δ*d* = 1.6 mm	A_0_ mode	
		137	34,000 1/s
		S_0_ mode	
		3	7500 1/s

**Table 5 sensors-22-06750-t005:** The uncertainty budget of the combined uncertainty of the reconstructed frequency range (114.3–213.6) kHz for the A_0_ mode. The shadows represent the dominant components.

Quantity	Value	Standard Uncertainty	Distribution	Sensitivity Coefficient	Uncertainty Contribution
$\Delta_{c_{ph}}$	9.85 m/s	1.67 m/s	normal	1.0	1.7 m/s
$\Delta_{c_{mod}}$	0.0 m/s	3.28 m/s	normal	1.0	3.3 m/s
$\Delta_{c_{phmax}}$	0.0 m/s	1.2 m/s	normal	1.0	1.2 m/s
$\Delta_{l}$	0.0 m	28.9 × 10^−6^ m	rectangular	600 × 10^3^ s^−1^	17 m/s
$\Delta_{d}$	0.0 m	289 × 10^−6^ m	rectangular	34 × 10^3^ s^−1^	9.9 m/s
$\Delta_{}$	0.0 kg/m^3^	0.289 kg/m^3^	rectangular	$0.2\;m^{4}/s\cdot \mathrm{kg}$	0.06 m/s
$\Delta_{}$	0.0 m	28.9 × 10^−6^	rectangular	200 m/s	5.9 × 10^−3^ m/s
$\Delta_{E}$	0.0 GPa	289 × 10^−6^ GPa	rectangular	7.3 m/s·GPa	2.1 × 10^−3^ m/s
$c_{ph}$	1599.8 m/s	20.3 m/s			
Result value:		Expanded uncertainty:	Coverage factor:		Coverage:
1600 m/s		±41 m/s	2.00		95% (normal)

**Table 6 sensors-22-06750-t006:** The uncertainty budget of the combined uncertainty of the reconstructed frequency range (137.5–186.5) kHz for the S_0_ mode. The shadows represent the dominant components.

Quantity	Value	Standard Uncertainty	Distribution	Sensitivity Coefficient	Uncertainty Contribution
$\Delta_{c_{ph}}$	−86.8 m/s	34.8 m/s	normal	1.0	35 m/s
$\Delta_{c_{mod}}$	0.0 m/s	5.79 m/s	normal	1.0	5.8 m/s
$\Delta_{c_{phmax}}$	0.0 m/s	19.1 m/s	normal	1.0	19 m/s
$\Delta_{l}$	0.0 m	28.9 × 10^−6^ m	rectangular	600 × 10^3^ s^−1^	17 m/s
$\Delta_{d}$	0.0 m	289 × 10^−6^ m	rectangular	7500 s^−1^	2.2 m/s
$\Delta_{}$	0.0 kg/m^3^	0.289 kg/m^3^	rectangular	$1.2\;m^{4}/s\cdot \mathrm{kg}$	0.33 m/s
$\Delta_{}$	0.0 m	28.9 × 10^−6^	rectangular	1700 m/s	0.05 m/s
$\Delta_{E}$	0.0 GPa	289 × 10^−6^ GPa	rectangular	40 m/s·GPa	0.01 m/s
$c_{ph}$	5315.2 m/s	43.7 m/s			
Result value:		Expanded uncertainty:	Coverage factor:		Coverage:
5315 m/s		±87 m/s	2.00		95% (normal)

## Data Availability

Not applicable.

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
