# Peer review of "Accuracy Assessment of the 2D-FFT Method Based on Peak Detection of the Spectrum Magnitude at the Particular Frequencies Using the Lamb Wave Signals"

_sensors, 2022, doi:10.3390/s22186750_

Round 1

Reviewer 1 Report (New Reviewer)

The work is well structured and presents analytically convincing results, confirmed by an experimental study. In my opinion it can be published as is.

Author Response

Thank you very much for your recognition of our research presented in this paper.

Reviewer 2 Report (New Reviewer)

The manuscript concerns the assessment of the accuracy of the 2D-FFT method applied to Ultrasonic lamb Waves. The manuscript contains enough materials to deserve publication in the review sensors. However, I found that some paragraphs are not clear and should be rewritten (rephrasing). I found unusual (to  me) the way the results of  the phase velocities are presented (see e.g., lines 439-440). The authors abuse of the term "proposed method" and sometimes it is not easy to follow and understand the meaning. For these reasons and for the detailed comments shown below, I recommend to amended the manuscript.

--P1.

*L18. remove "the" from "the additional..."

*L27. remove "a" from "a clearly ...zones".

*L39. the first--> the firsts

*L40. remove "the" from "the propagating...".

--P2.

*L47. remove "the" from "the special signal"

*L51. remove "The" from "The ultrasonic inspection"

*L58; You may use [7-9] instead of [7] [8] [9]

*L61. are emerging --> emerge

*L63. [9][10]-->[9-10]

*L71. [11][12]-->[11-12]

*L80-81.  which previous work? You should add a reference. I thnik you mean "in our previous work" [13].

--P3.

*L109. You may indicate the structure (organization) of your manuscript at the end of section 1.

*L112-113. "an image ..are obtained"--> "an image ..is obtained".

--P5.

**L171. The 200 mm--> A 200 mm

*L176. You may say some words about the meaning of B-scan images in a general manner.

*L188. detection peaks --> detection of peaks

--P6.

*L199. In the caption of fig 4 dots are in reality circles or opne circles. Same remark for other figures.

--P7.

*L223. You may mention that te 0.2% are obtained by dividing 3.28 by 1590 times 100;

* As the mean relative error... are obtained quite accurately, ..". Is it because these errors have been obtained accurately or because of their low values? You sentence means thatthe way you calculate the relative error and the standard deviation is accurate. Do you mean this?

*L230. remove "of the research".

--P8.

*L248-249 "aredetected the slight scattered". This is not clear. To rephrase.

*L249. remove "the" from "the estimating".

*L250. I should say "The studies carried out [18] on the S0 mode..."

--P9.

*L279-282. In the caption, theterm "proposed" method is repeated. It is not easy to follow... can you be more explicit? same remark for proposed algorithm.  (dots (c)) should read circles or open circles.

--P10

*L304. and presented --> described

*L305. is presented --> are presented

*L310. study is --> study was

*L312. these transducer-->these transducers

*L322. experiments are conducted --> experiments were conducted.

*L327-329. The mentioned distance are measured from which point? from the position of the transmitter ? Do you mean Delta x in figure 8?

--P11.

Fig. 9. y-axis. What does n.a. in the amplitude unit mean?

*L335. remove "having"

*L337. section title "proposed method". can you be more explicit?

*L340. Same remark as for the title of section 4.2

--P12.

*Fig 10 c. Indicate A_0 and S_0 mode for the two curves.

Explain why there is 9 points for the upper curve and 26 points for the lower one. (dots) are circles.

*L359. [20][21]-->[20-21]

*L367. Why the sum goes up to K=26?

*L376. mode-->modes. ", the standard deviation is"-->"and the standard deviation is"

--P13.

Table 3. Can you explain the meaning of a negative avlue of -86.84?

*L394. velocities errors-->velocity errors.

*L414. I think there is a mistake in relation (8). [F(x_i-Delta x-i)-F(x_i)] should be replaced by [F(x_i+Delta x-i)-F(x_i)].

*L421. "The input data .. is presented"--> "The input data ...are presented".

--P14.

*L426. In the table you may use mm instead of m to avoid 1E-3.

*L439-440. What do you mean by 1590m/s+0.6% +-2.6%? what does +0.6% mean? what does +/-2.6% mean? Same for other values.

--P15.

The quality and utility of fig 11 are not convincing.

*L450-451. The sentence is not clear. To rephrase.

*L455-466. The paragraph is not written in a clear manner. Please rephrase it especially lines 455-58 and line 460.

--P16.

*L474 "well" ?

*L483. "including the uncertainty" what does this mean?

*L488-489. See comment on page 14.

References.

The volumes and page numbers are missing in all the references.

Why-1.6% in  5402 m/s-1.6%?

Author Response

Reviewer 3 Report (New Reviewer)

This edition manuscript can be accepted.  Additionly, authors should highlights the potential application of this text in introduction and conclusions.

Author Response

Reviewer remarks

This edition manuscript can be accepted.  Additionly, authors should highlights the potential application of this text in introduction and conclusions.

Response of the authors

In the introduction was added:

“As presented in [12], not only the location of the defect, but also its size is determined according to the changes of the phase velocity. Therefore, algorithms allowing direct reconstruction of the quantitative values of the dispersion curves by (f,v) are constantly developed and researched. However, it should be noted that in order to estimate the location and size of the defect based on phase velocity changes, at first it is necessary to know the reliability of these methods. Therefore, an assessment of the accuracy of any method developed must be performed.”

In the conclusions was added:

“The obtained results of phase velocity measurement for the A0 mode in the range (114.3 – 213.6) kHz and for the S0 mode in the range (137.5 – 186.5) indicate that the proposed method is a reliable tool for the quantitative evaluation of the phase velocity dispersion values of Lamb waves. This facilitates the selection of the method according to the required parameters, solving one or another problem in various future studies.”

Reviewer 4 Report (New Reviewer)

In this paper, the 2D-FFT algorithm is applied to the peak detection of spectrum, and the processing architecture of the method is proposed. The feasibility and accuracy of the proposed method are verified by simulation and experiment using the RF signal of ultrasonic Lamb wave (ULW). It can be seen that the author has done a lot of tests to prove the feasibility of the method from different aspects under a number of indicators, and the workload is full. However, there are still some problems to be solved before acceptance:

1. This paper is more like a technical report with a lot of experiments, and the innovation is more about the experimental architecture than the algorithm. Should the article put more emphasis on experimental work?

2. Should equation (2) be the absolute value of the difference between the two parameters?

3. The signal adopted in this paper are Ultrasonic Lamb wave. Can it be assumed that this wave must be used when using the author's method, and are there any limitations in environmental setting and so on? What are the advantages and disadvantages of using other types of waves?

Author Response

In this paper, the 2D-FFT algorithm is applied to the peak detection of spectrum, and the processing architecture of the method is proposed. The feasibility and accuracy of the proposed method are verified by simulation and experiment using the RF signal of ultrasonic Lamb wave (ULW). It can be seen that the author has done a lot of tests to prove the feasibility of the method from different aspects under a number of indicators, and the workload is full. However, there are still some problems to be solved before acceptance:

Reviewer remarks

  1. This paper is more like a technical report with a lot of experiments, and the innovation is more about the experimental architecture than the algorithm. Should the article put more emphasis on experimental work?

Response of the authors

We would like to draw your attention that the experimental part is related to verification of our method and mainly discussed in one chapter of the article (Chapter 4), but not in a whole article. According to the requirements of ISO 17025, the method validation and verification are to be carried out by quantitative assessment of its accuracy. Therefore, the key sequence of the mathematical and experimental stages are necessary to be performed to validate the newly developed or adapted methods. That were done and presented in our work.

Reviewer remarks

  1. Should equation (2) be the absolute value of the difference between the two parameters?

Response of the authors

The absolute error is equal to the difference between the two velocities at one point corresponding to a certain frequency. One velocity is obtained according to the reference dispersion curve, and another to the mathematical simulation curve.

Reviewer remarks

  1. The signal adopted in this paper are Ultrasonic Lamb wave. Can it be assumed that this wave must be used when using the author's method, and are there any limitations in environmental setting and so on? What are the advantages and disadvantages of using other types of waves?

Response of the authors

Lamb waves are one of the types of Ultrasonic Guided Waves (UGWs) that propagates in plates. The UGW can propagate long distances along the large engineering structures and constructions based on metal and composite materials, e.g. to propagate tens of meters or more along it. In the case of pipelines inspection, it is possible to cover more than 100 m. Therefore, from the excitation region we could inspect quite large region of the structure due to long distance propagation of such waves. So, the proposed method can be used for all types of UGWs.

Using other types of waves, such as bulk longitudinal or shear waves commonly used in NDT (non-destructive testing) practise, it is able to propagate in sufficiently short distances, e.g. less than 1 meter. Also, it is possible to inspect region of the structure just below the active surface of transmitting-receiving transducers, and not possible to inspect large areas.

Round 2

Reviewer 4 Report (New Reviewer)

I would like to recommend publication of this manuscript in Sensors because all the comments have been carefully responsed.

This manuscript is a resubmission of an earlier submission. The following is a list of the peer review reports and author responses from that submission.

Round 1

Reviewer 1 Report

Dear Authors.
Thank you very much for your reply. In view of the arguments raised to my objections, I have to admit that the authors are right and therefore my decision is to recommend the publication of the article. I thank that they have included modifications in the text that help to focus the discussion and the scope of the article. In my opinion, there were not very clear in the first version. Hence my initial criticism that the work seemed more like a technical report than a paper. Congratulations. Best regards,
The reviewer

Reviewer 2 Report

In the resubmitted version of the manuscript, only some minor changes were introduced by the authors, which, to my mind, are insufficient for its acceptance.

In the response to the reviewers' comments, it was mentioned that "The presented article is ... about the evaluation of the reliability of the developed method". If the authors are intending to make a thorough and exhaustive study of the method reliability, show its "metrology" and demonstrate that "further testing or calculations will not enhance the quality of the predictive/ expected measurement" it is strongly recommended to consider a more complex and realistic problem, which is sufficiently closer to real-world engineering applications.

To be more precise, application and sensitivity analysis of the proposed dispersion curve evaluation approach to the case of broadband multimode signals (e.g., to those from papers https://doi.org/10.1515/teme-2017-0132, https://doi.org/10.1016/j.compstruct.2021.114178, etc.) must be studied. 

Moreover, experimental verification should be performed for a less extended plate, which is closer to realistic engineering structures, so that multiple wavepackets reflected from structural boundaries are also present. In many cases such long distances from the source (as it is in the current experiment, i.e., 178 mm away from the excitation point) might not be available. Therefore, it might not be possible to separate single modes as it is done in Figure 9, and 2D-FFT should be applied directly to the obtained signal. This issue should be also carefully addressed. 

Regarding the “metrological characteristics” mentioned in the paper, the authors are comparing the results of 2D-FFT approach applied to the experimental data with the values obtained with the computational package “DISPERSE”. The latter were evaluated for some idealized values of elastic modulus mentioned in the beginning of the subsection 3.1. However, is there any assurance that these values also correspond to the alluminium plate employed in the experiments?

The authors might also like to provide more details regarding the experimental setup. For instance, what particular type of the ultrasonic transducer was used (manufacturer, diameter of the contact area), what was the sampling frequency?

Reviewer 3 Report

1. The authors do not accurately describe the drawbacks in the 2D-FFT method. The mentioned points in the manuscript are not very convincing. For example.

- “As a result, a number of various signal processing algorithms are created and used for the quantitative display of (f, k) domains for the dispersion evaluation of Lamb waves.”

The 2D-FFT obtains a contour map about f-k, and the magnitude of each point is able to be analyzed quantitatively. 

- “However, the dispersion curves of such waves are more appropriate to be displayed by the plot of the frequency and velocity (f, v).”

Actually, the (f, cp) spectrum is one of the common analytical components of the 2D-FFT. The example of 2D-FFT mentioned by Joseph L Rose in his classic book "ultrasonic guided waves in solid media" uses the (f, cp) spectrum. Due to the simplicity of the (f, cp) spectrum acquisition, its not appropriate to be the main issue of 2D-FFT. 

- “Then the representation can be directly associated with the calculation of the velocity changes, which are used for detecting defects and/or delamination [12], and provide understanding about the propagation phenomena, such as: reflections, refractions, diffractions, mode conversions, and others [5].”

The analysis of reflections, refractions, and diffractions should focus on group velocity analysis rather than phase velocity.

2. Section 3 applies the numerical method to generate the wavefield signal, which is transformed to obtain the (f, cp) spectrum, and the results are compared with the dispersion curves calculated by DISPERSE. There is an important question here, the generated signal ux(t) depends on the phase velocity cph, and the obtained results must be in high consistency with the phase velocity dispersion curve of the Lamb wave, and the error of the analysis is only derived from the FFT. What is the significance of analyzing the A0 and S0 models separately? And the manuscript does not analyze the source of the error and how to improve it.

3. The receiver moves in increments of 0.1 mm. does it take into account the effect of sensor size on it?

4. In Section 4, why separate the A0 mode from the S0 mode in the time-distance domain? This is unnecessary work from the subsequent analysis process.

5. There is a high degree of overlap between the simulation and experimental studies. What is the significance of analyzing them separately?

6. The impact of the uncertainty of each indicator should be analyzed with specific cases. Otherwise, the content of Section 5 is more separated from the content of Sections 3 and 4.

7.Monte Carlo simulation is applied in the manuscript, how is it implemented? What is the purpose of applying it?